# Perioperative Management and Clinical Outcomes of Liver Transplantation for Children with Homozygous Familial Hypercholesterolemia

**DOI:** 10.3390/medicina58101430

**Published:** 2022-10-11

**Authors:** Huan-Rong Qiu, Liang Zhang, Zhi-Jun Zhu

**Affiliations:** 1Department of Anesthesiology, Beijing Friendship Hospital, Capital Medical University, Beijing 100050, China; 2Liver Transplantation Center, National Clinical Research Center for Digestive Diseases, Beijing Friendship Hospital, Capital Medical University, Beijing 100050, China

**Keywords:** homozygous familial hypercholesterolemia, perioperative management, liver transplantation, children, cardiovascular comorbidities, postoperative complications, survival

## Abstract

*Background and Objectives:* Liver transplantation (LT) has been accepted as a life-saving option as a last resort for children with homozygous familial hypercholesterolemia (HoFH). Perioperative management of LT for HoFH poses extra challenges for clinicians largely due to premature atherosclerotic cardiovascular diseases (ASCVDs). We aimed to analyze our data of pediatric LT recipients with HoFH, with special attention paid to perioperative management and clinical outcomes. *Materials and Methods*: After obtaining approval from the local ethics committee, the clinical data of pediatric patients with HoFH who underwent LT at our institution between January 2014 and February 2021 were retrospectively studied. *Results*: Six pediatric LT recipients with HoFH were included in the analysis. Although ASCVDs were common before LT, all children with HoFH survived the perioperative period without in-hospital mortality. However, one patient experienced acute myocardial infarction two months following LT and was successfully treated with medical interventions. Post-LT metabolic improvement was shown by declines in serum total cholesterol (TC) and low-density lipoprotein cholesterol (LDL-C) levels in the early post-LT period (for TC: 14.7 ± 3.2 mmol/L vs. 5.5 ± 1.8 mmol/L, *p* < 0.001; for LDL-C: 10.6 ± 2.2 mmol/L vs. 3.6 ± 1.2 mmol/L, *p* < 0.001, respectively) and at the last follow-up (for TC: 14.7 ± 3.2 mmol/L vs. 4.5 ± 0.9 mmol/L, *p* = 0.001; for LDL-C: 10.6 ± 2.2 mmol/L vs. 2.8 ± 0.6 mmol/L, *p* = 0.001, respectively). Dietary restrictions could be lifted after LT. However, three patients required restarting lipid-lowering therapy after LT due to suboptimal LDL-C levels and progression of ASCVDs. *Conclusions*: Our data suggest that LT can be a safe and feasible therapeutic option for well-selected patients with HoFH, offering relaxed dietary restrictions and remarkable reductions in LDL-C levels. However, concerns remain regarding progression of ASCVDs after LT.

## 1. Introduction

Familial hypercholesterolemia (FH) is an autosomal dominant inherited metabolic disorder caused by mutations in low-density lipoprotein (LDL) receptor (LDLR), apolipoprotein B-100 (Apo B), proprotein convertase subtilisin/kexin type 9 (PCSK9), or LDLR adaptor protein 1 (LDLRAP1) genes. FH is generally classified into two phenotypes: heterozygous FH (HeFH) and homozygous FH (HoFH). The former is relatively common, with a reported prevalence of 1:500, whereas the latter is scarce, with an approximate incidence of 1:1,000,000 [1,2,3]. In both phenotypes, FH is characterized by elevated LDL cholesterol (LDL-C) levels, xanthomatosis, and premature atherosclerotic cardiovascular diseases (ASCVDs). However, patients with HoFH have been reported to have remarkably higher LDL-C levels with earlier onset and more accelerated progression of ASCVDs than those with HeFH [1,2]. If left untreated, patients with HoFH typically develop aortic valve and coronary artery disease (CAD) and even experience cardiac death in the second or third decade of life [2,3,4,5,6,7].

Conventional therapeutic options, including statins with or without ezetimibe [8,9,10,11] and LDL apheresis [12,13], are usually insufficient to achieve ideal LDL-C treatment targets and prevent cardiovascular consequences in HoFH patients. The latest approved novel lipid-lowering agents, such as lomitapide [14,15], mipomersen [16,17], PCSK9 inhibitors [18,19,20,21,22,23], and evinacumab [24,25], are significant advances. However, their use is limited by availability, cost, and age restrictions. Other promising approaches, such as gene therapy [26,27] and hepatocyte transplantation [28], are currently being explored. Finally, liver transplantation (LT) has been considered a therapeutic alternative for HoFH patients, especially when all other traditional options have been exhausted [29,30,31,32,33,34,35,36,37].

Perioperative management of patients with HoFH is always challenging, primarily due to premature ASCVDs. However, there are only a few case reports regarding perioperative management of HoFH patients undergoing either dental, endovascular interventional, or open-heart surgery [38,39,40]. LT is generally accompanied by severe intraoperative hemodynamic disturbances, making it one of the highest-risk noncardiac surgeries. To our knowledge, no study has focused on perioperative issues of LT for HoFH patients. Thus, we report our perioperative experience with a case series of HoFH children treated with LT.

## 2. Materials and Methods

### 2.1. Patients

This study was initiated after obtaining approval from the Institutional Review Board of Beijing Friendship Hospital (No.2021-P2-080-01). Pediatric patients (younger than 18 years) with a diagnosis of HoFH who underwent LT between January 2014 and February 2021 at Beijing Friendship Hospital were included in the study. HoFH was defined by meeting at least one of the following criteria: (1) genetic confirmation of two mutant alleles of LDLR, Apo B, PCSK9, and LDLRAP1 gene locus or (2) an untreated LDL-C > 13 mmol/L or treated LDL-C ≥ 8 mmol/L together with presence of xanthomas before the age of 10 years or diagnosis of HeFH in both parents [4].

### 2.2. Preoperative Evaluation

In our clinical practice, patients were generally screened for clinical presentations, medical history, family history, physical examinations, laboratory tests, and imaging detection (e.g., echocardiography, coronary computed tomography, or carotid vascular ultrasound) to reach a diagnosis of HoFH. Furthermore, special attention was paid to preoperative medical treatments, dietary restrictions, and disease-specific ASCVDs. For patients with a prior history of CAD, valvular heart disease, stroke, or who were noted to have severe cardiovascular comorbidities complicated by cardiac insufficiency or pulmonary hypertension, a multidisciplinary discussion was arranged to assess the appropriateness and optimal timing of LT.

### 2.3. Anesthesia and Surgery

The anesthetic and surgical techniques were performed according to the standard protocols of our institution. Briefly, anesthesia was induced with propofol, fentanyl, and cisatracurium and was maintained on inhalational and intravenous anesthesia with sevoflurane, remifentanil, and cisatracurium. Mechanical ventilation was initiated at a tidal volume of 6–8 mL/kg, a frequency of 12–20/min, and an FiO_2_ of 0.6. An arterial line was inserted into the radial artery for continuous monitoring of invasive arterial pressure, and a triple-lumen central venous catheter was inserted into the right internal jugular vein for continuous monitoring of central venous pressure and fluid and vasopressor infusions. A Swan–Ganz catheter was placed into the pulmonary artery in selected high-risk cases (e.g., left ventricular insufficiency, pulmonary hypertension, recent myocardial infarction, or severe ischemic heart disease) to measure cardiac output, pulmonary arterial pressure, and mixed venous oxygen saturation. All the liver grafts were procured from China Donation after Citizen’s Death donors and were preserved with the University of Wisconsin solution. Anastomosis of the liver graft was performed using a conventional caval replacement technique. Although the inferior vena cava was totally clamped during the anhepatic stage, neither temporary portocaval shunt nor venovenous bypass was used for any patient. Intraoperative management was mainly focused on correcting hemodynamic disturbances and maintaining the balance between oxygen delivery and oxygen consumption, especially during critical periods, such as anesthesia induction, inferior vena cava clamping, and graft reperfusion. After surgery, all patients were transferred to the intensive care unit for postoperative care.

### 2.4. Immunosuppression and Follow-Up Protocol

The basic immunosuppressive regimen consisted of methylprednisolone and tacrolimus. Treatment with intravenous methylprednisolone (10 mg/kg) was initiated before reperfusion. Intravenous methylprednisolone was gradually tapered from the recommended initial dose of 3–4 mg/kg/day during the first week postoperatively and was switched to oral methylprednisone at 1 mg/kg/day on postoperative day 8. Oral tacrolimus was administered on postoperative day 1 at an initial dose of 0.1–0.2 mg/kg/day. The target level of tacrolimus was 8–10 ng/mL. In addition, treatment with 10 mg of intravenous basiliximab was used in children weighing at least 20 kg before reperfusion and on postoperative day 4. Postoperative lipid profiles were monitored daily or bidaily until first discharge and monthly, quarterly, or semiannually thereafter, depending on lipid profile status. Carotid vascular ultrasound, echocardiography, and coronary computed tomography were performed semiannually, annually, or every three years, depending on vascular and valvular status.

### 2.5. Data Collection and Statistical Analysis

The following data were recorded: age, sex, weight, height, medical history, family history, graft type, graft weight, graft-to-recipient weight ratio, cold ischemia time, warm ischemia time, operation time, intraoperative blood loss, urine output, total fluid infusion, blood transfusions, occurrence of postreperfusion syndrome [41], acute myocardial infarction (AMI) [42], and in-hospital death. The other variables collected included diet restriction, LDL-C and total cholesterol (TC) levels, and echocardiography, coronary computed tomography, and carotid vascular ultrasound findings preoperatively, postoperatively, and at final follow-up. Values are represented as the mean ± standard deviation (SD), median and interquartile range (IQR), or number and proportion. Paired observations of TC and LDL-C levels at baseline, during the early post-LT period, and at the last follow-up were assessed with the Wilcoxon signed-rank sum test or paired *t*-test, as appropriate. Differences were considered significant if the *p* value was < 0.05. Statistical analyses were performed using SPSS for Windows software (version 22.0; IBM SPSS, Inc., Chicago, IL, USA).

## 3. Results

### 3.1. Preoperative Characteristics and Clinical Course

During the study period, six children underwent LT for HoFH at our institution. All the patients met the phenotypic and genetic criteria for HoFH before admission to our institution and were transplanted due to refractory hypercholesterolemia (i.e., occurrence of insensitivity to existing lipid-lowering therapy together with LDL-C levels ≥ 3.5 mol/L) [4]. At the time of LT, their mean age was 7 years (range, 2–12 years), 83.3% were male, and they had a mean height of 124 cm (range, 90–146 cm) and a mean weight of 23.8 kg (range, 14.0–30.0 kg) (Table 1). All patients were treated with a low-saturated-fat, low-cholesterol diet, five of whom had early signs of ASCVDs. The most frequently used lipid-lowering medications were statins (*n* = 4) and ezetimibe (*n* = 4), followed by probucol (*n* = 2) (Table 2).

### 3.2. Intraoperative Parameters and Clinical Course

All the liver grafts were whole grafts. The mean graft weight was 518.3 ± 76.5 g, and the mean graft-to-recipient weight ratio was 2.48 ± 1.10%. The average cold and warm ischemia times were 525 ± 150 min and 4 ± 1 min, respectively. The median (IQR) blood loss was 4.2 (2.7) mL/kg, and the mean urine output and fluid infusion were 2.7 ± 1.3 mL/kg and 84.3 ± 37.7 mL/kg, respectively. Nontransfusion was attained in four (66.7%) patients, and fresh frozen plasma was only indicated for one (16.7%) patient. The incidence of postreperfusion syndrome was 66.7%, but no patients experienced ECG signs of acute myocardial ischemia, heart failure, or cardiac arrest. Finally, all patients survived LT without in-hospital mortality (Table 1).

### 3.3. Postoperative Complications and Clinical Outcomes

Patient #3 experienced AMI two months after LT and was successfully treated with medical interventions. Patient #4 developed hepatic artery thrombus without elevated liver enzymes on postoperative day 11, and he had received oral warfarin and intravenous prostaglandin E1 for anticoagulation. Finally, the patient was noted to have arterial collateral formation on postoperative day 19; he did not suffer graft failure or biliary complications in the post-LT period. Patient #5 was successfully treated with increased immunosuppression for an acute rejection episode that occurred 10 days after LT (Table 1). At one month post-LT, the dietary cholesterol restriction was lifted in all HoFH children (Table 2). The postoperative TC and LDL-C levels decreased significantly during the early post-LT period (for TC: 14.7 ± 3.2 mmol/L vs. 5.5 ± 1.8 mmol/L, *p* < 0.001; for LDL-C: 10.6 ± 2.2 mmol/L vs. 3.6 ± 1.2 mmol/L, *p* < 0.001, respectively) and at the time of the last follow-up (for TC: 14.7 ± 3.2 mmol/L vs. 4.5 ± 0.9 mmol/L, *p* = 0.001; for LDL-C: 10.6 ± 2.2 mmol/L vs. 2.8 ± 0.6 mmol/L, *p* = 0.001, respectively) when compared with the baseline levels (Figure 1). Despite this, patient #5 did not achieve an ideal LDL-C level of < 3.5 mmol/L after LT, and two patients (patients #1 and #3) were noted to have ASCVD progression at the last follow-up (Table 2). Consequently, these patients still needed to be treated with lipid-lowering agents. Until the last follow-up date of 31 August 2022, graft and patient survival were 100% (Table 2).

## 4. Discussion

Little is known about the perioperative issues of LT for patients with HoFH. Herein, we describe 100% patient and graft survival in six HoFH children with a mean follow-up of forty-six months. Despite the small case series, several clinical implications can be drawn: (1) children with HoFH are often complicated with varying degrees of ASCVDs before LT, which may increase the risk of adverse cardiovascular events after LT. (2) LT can significantly reduce LDL-C levels and lift dietary restriction in children with HoFH. (3) Risk of progression of ASCVDs in HoFH children is not entirely eliminated after LT.

HoFH is a rare autosomal dominant disorder that typically manifests as premature ASCVDs owing to markedly elevated LDL-C levels. ASCVDs in HoFH patients include CAD, hypercholesterolemic valvulopathy, and carotid, aortic, renal, and peripheral artery stenosis. If not diagnosed and treated early, patients with HoFH often develop severe ASCVDs and may experience sudden cardiovascular death from progressive CAD or aortic valve disease in the second or third decade of life [2,3,4,5,6,7]. Unfortunately, HoFH is underdiagnosed and undertreated in developed and developing countries [1,2]. Diagnosis of HoFH can usually be made using a combination of clinical findings, laboratory results, family history, and genetic testing. According to the current diagnostic criteria endorsed by the European Atherosclerosis Society [4], HoFH is diagnosed based on either phenotypic or genetic criteria. The phenotypic criteria of HoFH typically include an untreated LDL-C level of > 13 mmol/L or a treated LDL-C level of ≥ 8 mmol/L with the presence of xanthomas before the age of 10 years or evidence of HeFH in both parents. The genetic diagnosis of HoFH requires identification of mutations in two alleles at gene loci for LDLR, Apo B, PCSK9, or LDLRAP1.

The current guidelines recommend that HoFH patients receive aggressive lipid-lowering regimens as soon as possible after diagnosis [3,4,6] because the risk of ASCVDs increases with age and decreases with decreasing LDL-C levels [43,44]. Statins, either alone or in combination with ezetimibe, remain the cornerstone of the medical treatment of HoFH in the pediatric population [8,9,10,11]. Probucol was one of the leading pharmacologic options for HoFH before the era of statins but is now only available in some Asian countries because of prolongation of the QT interval and reduction in high-density lipoprotein cholesterol [45]. Other novel lipid-lowering agents, including lomitapide [14,15], mipomersen [16,17], PCSK9 inhibitors (e.g., alirocumab, evolocumab, and inclisiran) [18,19,20,21,22,23], and evinacumab [24,25], have been shown to significantly reduce LDL-C levels in adult HoFH patients. However, only mipomersen, evolocumab, and evinacumab have been approved by the US Food and Drug Administration as adjunct therapies for HoFH in patients aged 12 years and older [46]. Despite these lipid-lowering agents, guideline-recommended LDL-C targets of less than 3.5 mmol/L are not achieved in some pediatric patients with HoFH; therefore, alternative therapies are urgently needed. The current standard of care for pediatric patients with HoFH is LDL apheresis [12,13], which reduces LDL-C levels dramatically. However, it fails to halt progression of ASCVDs as LDL-C levels often remain above acceptable targets. In addition, this therapy is costly, invasive, and time-consuming for pediatric HoFH patients. Other attractive approaches, such as gene therapy [26,27] or hepatocyte transplantation [28], are largely underexplored. Finally, LT has been a last resort for pediatric patients with HoFH since the 1980s [29,30,31,32,33,34,35,36,37]. In our case series, none of these six children with HoFH received aggressive lipid-lowering therapy with mipomersen, evolocumab, evinacumab, or LDL apheresis before LT, which may reflect a real problem with undertreatment in settings where availability of lipid-lowering therapy is limited.

Theoretically, LT enables replacement of intrahepatic LDL receptors, which account for approximately 75% of the total dysfunctional LDL receptors [37]; therefore, LT offers a potentially curative option for HoFH, but its use is limited by organ scarcity, economic reasons, transplant-related morbidity and mortality, and the requirement of lifelong immunosuppression. Consequently, LT is not generally recommended as the first-line treatment of HoFH [4,47]. Although substantially lowering the LDL-C level may be achieved with LT, the reduction in LDL-C levels may fluctuate and rebound, and the progression of ASCVDs can be problematic. In line with our results, previous studies have reported fluctuating LDL-C levels in HoFH patients after LT [30,31,32,33,34,37], perhaps because of residual extrahepatic LDL receptors and immunosuppressive drug-induced hyperlipidemia [37]. Consequently, postoperative lipid-lowering therapy may be crucial for achieving acceptable LDL-C levels and halting progression of ASCVDs. Unfortunately, the optimal LDL-C targets for pediatric LT recipients with HoFH have not been determined. Based on guideline-recommended LDL-C targets in HoFH children, it seems reasonable to reach a target LDL-C goal of < 3.5 mmol/L following LT [4,5,6,7]. Consistent with prior studies [36,48], our study indicated that ASCVDs in children with HoFH could be exacerbated after LT, even when existing LDL-C goals are attained. However, whether a stricter target LDL-C level would reduce the risk of ASCVD progression following LT remains unclear, and research evidence is limited.

Perioperative management of LT for children with HoFH represents a medical challenge for clinicians. The complexity of these challenging patients and the specific risks of LT require the collaboration of a multidisciplinary team that consists of LT surgeons, anesthesiologists, hepatologists, cardiologists, cardiac surgeons, neurological physicians, pediatricians, intensivists, and radiologists. Generally, patients with HoFH are reported to have severe cardiovascular comorbidities, such as CAD and aortic valve disease, which sometimes require surgical and nonsurgical interventions [38,39,40,49]. Therefore, caution must be taken in assessing cardiovascular risk to ensure patient safety since preoperative ASCVDs will inevitably increase the risk of postoperative cardiovascular complications [32,33,37]. In our series, one patient experienced AMI in the early postoperative period despite careful cardiac precautions, suggesting an elevated cardiac risk in HoFH patients treated with LT. In addition, it should be noted that pediatric HoFH patients must be strictly followed up because of the suboptimal LDL-C levels and progression of ASCVDs after LT. Based on our experience, we have established a perioperative management algorithm for HoFH children treated with LT (see Table 3).

To our knowledge, this is the first study mainly focusing on perioperative issues in HoFH children treated with LT. Other strengths include the representative nature of the patient sample, the detailed information on laboratory and imaging results, and the high response rate during follow-up. Nevertheless, several limitations also deserve mention. First, this study is limited by the relatively small sample size and lack of a control group; thus, future studies will be necessary to confirm our observations. Second, most children with HoFH did not receive guideline-recommended lipid-lowering agents because of limited drug availability, side effects, or high cost. Future studies are warranted to determine whether aggressive lipid-lowering therapy can achieve acceptable LDL levels to avoid the possibility of LT. Third, to minimize the risk of ASCVD progression, the optimal LDL-C level in HoFH children after LT remains to be determined. Finally, the median follow-up was relatively short; therefore, long-term graft and patient survival should be observed.

## 5. Conclusions

Our data suggest that LT can be a safe and feasible therapeutic option for well-selected children with HoFH, offering relaxed dietary restrictions and remarkable reductions in LDL-C levels. However, it is essential to recognize that some HoFH children treated with LT still require reinitiation of lipid-lowering therapy following LT because of suboptimal LDL-C levels and ASCVD progression. Therefore, adequate preoperative evaluation, meticulous perioperative management, and strict postoperative follow-up are critical to optimizing clinical outcomes of LT for children with HoFH.

## Figures and Tables

**Figure 1 medicina-58-01430-f001:**
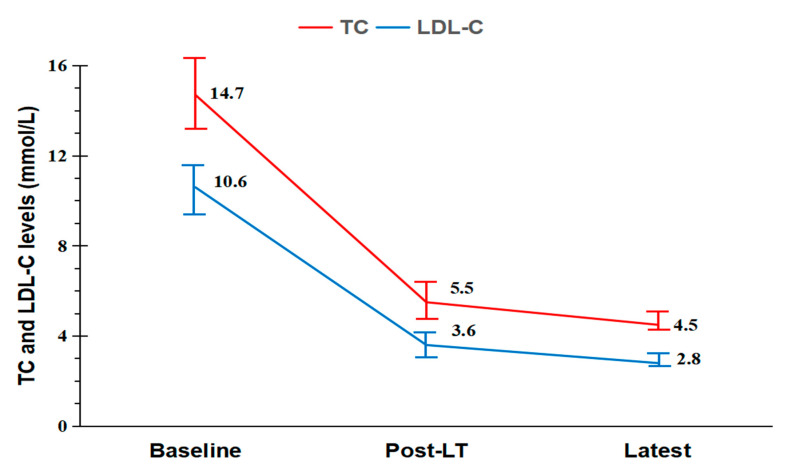
Changes in serum TC and LDL-C levels over time. Baseline levels were obtained in the immediate preoperative period. Post-LT levels were defined as those measured at normalization, first discharge, or one month following LT, whichever occurred first. The latest levels were obtained at the time of the last follow-up. Significant decreases in postoperative serum TC and LDL-C levels were noted. Post-LT TC levels: mean 5.5 mmol/L versus baseline levels: mean 14.7 mmol/L; *p* < 0.001. Latest TC levels: mean 4.5 mmol/L versus baseline levels: mean 14.7 mmol/L; *p* = 0.001. Post-LT LDL-C levels: mean 3.6 mmol/L versus baseline levels: mean 10.6 mmol/L; *p* < 0.001. Latest LDL-C levels: mean 2.8 mmol/L versus baseline levels: mean 10.6 mmol/L; *p* = 0.001. LDL-C low-density lipoprotein cholesterol, LT liver transplantation, TC total cholesterol.

**Table 1 medicina-58-01430-t001:** Baseline characteristics, intraoperative parameters, and related complications.

	Case 1	Case 2	Case 3	Case 4	Case 5	Case 6
Age at onset (year)	6	1	3	2	1	1
Age at admission (year)	12	1	9	7	5	3
Age at LT (year)	12	2	10	7	6	3
Sex	Male	Female	Male	Male	Male	Male
Weight at LT (kg)	30.0	12.0	26.0	28.0	28.0	16.5
Height at LT (cm)	146	90	144	134	115	106
Xanthomas	Yes	Yes	Yes	Yes	Yes	Yes
ASCVDs	Yes	No	Yes	Yes	Yes	Yes
Family history	FDR: HeFH	No	FDR: HeFH; SDR: Premature CAD	FDR: HeFH, Premature CAD	FDR: HeFH; SDR: HeFH	SDR: Premature CAD
Cardiac history	No	No	Angina	CAD	No	Myocarditis
Graft weight (g)	568	540	571	575	383	473
GRWR (%)	1.89	4.50	2.20	2.05	1.37	2.87
Warm ischemia time (min)	5	5	3	5	3	3
Cold ischemia time (min)	520	720	420	560	630	300
Operation time (min)	461	312	224	145	380	205
Blood loss (mL/kg)	3.3	4.2	3.8	5.0	4.2	10.3
Urine output (mL/kg/h)	4.9	3.2	2.2	2.1	1.0	2.9
Fluid infusion (mL/kg)	152.5	62.5	85.7	42.9	69.6	92.3
RBC (mL/kg)	8.7	21.7	0	0	0	0
FFP (mL/kg)	6.7	0	0	0	0	0
Complications	PRS	No	AMI	PRS, HAT	PRS, rejection	PRS

ASCVD, atherosclerotic cardiovascular disease; AMI, acute myocardial infarction; CAD, coronary artery disease; FDR, first-degree relatives; FFP, fresh frozen plasma; GRWR, graft-to-recipient weight ratio; HAT, hepatic artery thrombosis; HeFH, heterozygous familial hypercholesterolemia; LT, liver transplantation; PRS, postreperfusion syndrome; RBC, red blood cell; SDR, second-degree relatives.

**Table 2 medicina-58-01430-t002:** Lipid profiles, ASCVDs, diet restriction, and lipid-lowering therapy before and after LT.

	Case 1	Case 2	Case 3	Case 4	Case 5	Case 6
Baseline TC (mmol/L)	17.48	14.06	12.66	12.72	11.63	19.63
Post-LT TC (mmol/L)	7.03	7.67	5.08	3.53	3.47	6.27
Latest TC (mmol/L)	4.00	4.61	4.19	4.04	6.27	3.94
Baseline LDL-C (mmol/L)	13.49	10.11	8.99	8.96	8.81	13.23
Post-LT LDL-C (mmol/L)	5.02	4.68	3.40	2.07	2.43	4.15
Latest LDL-C (mmol/L)	2.40	2.81	2.70	2.50	4.02	2.51
Pre-LT CVUS	cIMT↑, CAS (moderate)	Normal	cIMT↑	cIMT↑, CP	cIMT↑	cIMT↑
Latest CVUS	cIMT↑	Normal	CAS (severe)	cIMT↑, CP	Normal	Normal
Pre-LT ECHO	SVAS (moderate)	Normal	Normal	Normal	Normal	Normal
Latest ECHO	AS (severe)	Normal	Left ventricular SWMA	Normal	Normal	Normal
Pre-LT coronary CT	RCA plaques (no stenosis)	Normal	RCA (mild stenosis)	Normal	Normal	Normal
Latest coronary CT	Normal	Normal	LM, LAD, LCX (severe stenosis); RCA (mild stenosis)	Normal	Normal	Normal
Change in ASCVDs	Exacerbated	Stabilized	Exacerbated	Stabilized	Stabilized	Stabilized
Pre-LT diet restriction	Yes	Yes	Yes	Yes	Yes	Yes
Post-LT diet restriction	Lifted	Lifted	Lifted	Lifted	Lifted	Lifted
Pre-LT lipid-lowering therapy	Statins	Statins	Statins, ezetimibe	Statins, ezetimibe, probucol	Ezetimibe	Ezetimibe, probucol
Latest lipid-lowering therapy	Statins	No	Statins	No	Ezetimibe	No
Follow-up (months)	93	79	37	23	25	19
Outcome	Alive	Alive	Alive	Alive	Alive	Alive

AS, aortic stenosis; ASCVD, atherosclerotic cardiovascular disease; CAS, carotid artery stenosis; cIMT, carotid intima-media thickness; CP, carotid plaque; CT, computed tomography; CVUS, carotid vascular ultrasound; ECHO, echocardiogram; LAD, left anterior descending coronary artery; LCX, left circumflex coronary artery; LDL-C, low-density lipoprotein cholesterol; LM, left main coronary artery; LT, liver transplantation; RCA, right coronary artery; SVAS, supravalvular aortic stenosis; SWMA, segmental wall motion abnormalities; TC, total cholesterol.

**Table 3 medicina-58-01430-t003:** Summary of perioperative considerations of LT for HoFH children.

	Considerations
**Preoperative**	
Diet restriction	A low-saturated-fat, low-cholesterol diet is necessary
Lipid-lowering therapy	Consider aggressive lipid-lowering therapy with lipid-lowering agents and LDL apheresis
Preoperative evaluation	ECHO, coronary CT, and CVUS can be used to assess the severity of ASCVDs thoroughly
Fasting	Prolonged fasting should be avoided
Premedication	Premedication is recommended, especially in crying and anxious children
**Intraoperative**	
Vascular access	Difficult vascular access may be encountered due to diffuse atherosclerosis
Anesthesia induction	Anesthesia can be induced under the guidance of IBP to avoid hemodynamic disturbances
Anesthesia maintenance	Ensure adequate depth of anesthesia
Monitoring	ECG, TEE, and NIRS can be used to detect any sign of myocardial and cerebral ischemia
Management	Maintain the balance between myocardial oxygen supply and demand
**Postoperative**	
Postoperative analgesia	Multimodal analgesia is recommended
Early monitoring	ECG, BNP, myocardial enzymes, and lipid profiles should be closely monitored
Immunosuppression	Tacrolimus-based immunosuppressive regimen in combination with methylprednisolone
Lipid-lowering therapy	Reinitiation of lipid-lowering therapy may be needed
Long-term follow-up	Focus on the progression of ASCVDs

ASCVD, atherosclerotic cardiovascular disease; BNP, brain natriuretic peptide; CT, computed tomography; CVUS, carotid vascular ultrasound; ECG, electrocardiograph; ECHO, echocardiogram; HoFH, homozygous familial hypercholesterolemia; IBP, invasive blood pressure; LDL, low-density lipoprotein; LT, liver transplantation; NIRS, near-infrared reflectance spectroscopy; TEE, transesophageal echocardiography.

## Data Availability

The data collected and analyzed during the current study are available from the corresponding author upon reasonable request.

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
