# Peer review of "Perioperative Management and Clinical Outcomes of Liver Transplantation for Children with Homozygous Familial Hypercholesterolemia"

_medicina, 2022, doi:10.3390/medicina58101430_

Round 1

Reviewer 1 Report

I thank you for the opportunity to go comment this manuscript.

Title

Shorten the title

Abstract

Authors conclusion is the following: “Conclusions: LT is a challenging procedure that may potentially provide a treatment strategy for changing the natural history of HoFH in well-selected cases, although its clinical outcomes are sometimes unfavorable due to its limited efficacy in achieving LDL-C targets and preventing atherosclerosis.”

I think that conclusion needs to also relate to perioperative management. Please, modify.

Intro and discussion

Authors mention only part of the medications used for the treatment of HoFH.

Check some comprehensive HoFH review and treatment guideline. Add text in intro and discussion. Try to build more balanced picture of the LDL pheresis and medications and their advances. Be aware also new very recent trials.

 Discuss also why the LLT treatment is important after the transplantation. 

I didn’t find any discussion related to issue that transplantation aggravates atherosclerosis due to medications need to be used. 

I wish that critical discussion is related to Xuezhikang. It needs to become clear that this medication is not recommended as a part of treatment. 

Authors need to also mention and be aware the limited use probucol worldwide. 

The cost-effective estimation needs to be discussed regarding transplantation of liver.

HoFH is not monotonous disease – discuss the different mutations and varying phenotype and prognosis.

Results

Divide table 1 into two different tables (operated non operated). 

Regarding table 1. Provide more detail ECHO, CVUS and Coronary CT. Was the Coronary CT really normal in the majority of the patients? I am surprised. Discuss this issue. I suggest that authors make new table describing these findings in detail. The preoperative valvular and arterial changes needs to be discussed in detail and literature (if not available from children then use adult literature) needs to be added.

Table 1. Add the age at diagnosis and age when the LLT initiated.

Table 2 can be appendix (and only shortened table can be part of the text).

Make additional table and carry out follow-up comparison of those HoFH patient having transplantation and those who didn’t. Discuss the prognosis both cases. Where there attempts to improve treatment among those who didn’t manage to get liver transplantation?

Figure 1. I think that readers are interested to get exact TC and LDL values. Provide table or make them otherwise available in the figure.

Figure 2. Provide also respective values for non-operated patients. Authors may like to know how fast LDL did decrease?

Additional tip: Because the journal is online you can use colours in figures. This will clarify them.

Language: some expressions are not so commonly used. Check the language.

I noticed that authors need to carry out carefully literature review and references. see also my earlier comments. Below is just as an example one important reference which is missing.

Additional references

Gao M, Yu W, Hu H, Liu H, Fan K, Gu C, Wang L, Yu Y. Case Report: Cardiac Surgery and Combined Lipid-Lowering Drug Therapy for Homozygous Familial Hypercholesterolemia. Front Pediatr. 2020 Oct 22;8:535949. doi: 10.3389/fped.2020.535949. PMID: 33194883; PMCID: PMC7642436.

Author Response

Replies to Reviewer 1

First of all, we would like to express our sincere gratitude to you for your constructive comments and hard work on our manuscript.

1. Title: Shorten the title.

Response: As suggested by the reviewer, we have changed the title of this manuscript to “Perioperative management and clinical outcomes of liver transplantation for children with homozygous familial hypercholesterolemia”.

2. Abstract: Authors conclusion is the following: “Conclusions: LT is a challenging procedure that may potentially provide a treatment strategy for changing the natural history of HoFH in well-selected cases, although its clinical outcomes are sometimes unfavorable due to its limited efficacy in achieving LDL-C targets and preventing atherosclerosis.”I think that conclusion needs to also relate to perioperative management. Please, modify.

Response: Thanks for your insightful suggestion. Accordingly, we have adjusted the inappropriate descriptions in the Abstract and Conclusions sections of the revised manuscript (Line 30–33, 312–319).

3. Intro and discussion: Authors mention only part of the medications used for the treatment of HoFH.Check some comprehensive HoFH review and treatment guideline. Add text in intro and discussion. Try to build more balanced picture of the LDL pheresis and medications and their advances. Be aware also new very recent trials.

Response: We are grateful for your valuable comments. Accordingly, we have summarized the advantages and limitations of various lipid-lowering drugs, LDL apheresis, gene therapy, hepatocyte transplantation and liver transplantation in the treatment of HoFH in detail in the Introduction and Discussion sections of the revised manuscript (Line 52–61, 231–256).

4. Discuss also why the LLT treatment is important after the transplantation. I didn’t find any discussion related to issue that transplantation aggravates atherosclerosis due to medications need to be used. The cost-effective estimation needs to be discussed regarding transplantation of liver.

Response: We quite agree with you that aggressive LLT after liver transplantation might be essential to reach guideline-recommended LDL-C targets and prevent the progression of atherosclerosis. As suggested by the reviewer, we have discussed about post-LT LLT and atherosclerosis progression in the Discussion section of the revised manuscript (Line 268–275). In addition, the disadvantages of LT for HoFH have also been added in the revised manuscript (Line 259–261).

5. I wish that critical discussion is related to Xuezhikang. It needs to become clear that this medication is not recommended as a part of treatment.

Response: Current treatment guidelines have not recommended Xuezhikang as a part of treatment for HoFH. The non-standard use of Xuezhikang may reflect a real problem with undertreatment in settings where the availability of lipid-lowering therapy limited. However, data on efficacy and safety of Xuezhikang in HoFH is unclear, particularly in the pediatric population. Therefore, the use of Xuezhikang was discontinued when the patient was admitted to our hospital for liver transplantation.

6. Authors need to also mention and be aware the limited use probucol worldwide. 

Response: The disadvantages of probucol for HoFH have been added in the Discussion section of the revised manuscript (Line 236–238).

7. HoFH is not monotonous disease – discuss the different mutations and varying phenotype and prognosis.

Response: Thank you for your precious comments and advice. Accordingly, we have discussed about the different phenotype and prognosis of FH patients in the Introduction section of the revised manuscript (Line 41–51).

8. Results: Divide table 1 into two different tables (operated non operated). 

Regarding table 1. Provide more detail ECHO, CVUS and Coronary CT. Was the Coronary CT really normal in the majority of the patients? I am surprised. Discuss this issue. I suggest that authors make new table describing these findings in detail.

Response: Considering the comments of other reviewers, only patients with HoFH who underwent LT were ultimately included in this study because the topic of this paper is LT for HoFH. As shown in Table 2 of the revised manuscript, we have provided more details of ECHO, CVUS and Coronary CT findings before and after LT, and have identified two patients with varying degrees of coronary stenosis before LT.

9. The preoperative valvular and arterial changes needs to be discussed in detail and literature (if not available from children then use adult literature) needs to be added.

Response: As you requested, we have discussed about the changes in preoperative ASCVDs in the Discussin section of the revised manuscript (Line 268–275).

10. Table 1. Add the age at diagnosis and age when the LLT initiated. Table 2 can be appendix (and only shortened table can be part of the text).

Response: We are extremely grateful to reviewer for pointing this out. However, since HoFH patients are generally underdiagnosed and undertreated in our country, we cannot provide the exact time when the LLT initiated in these patients. Despite of this, we have added the age at onset in Table 1 of the revised manuscript.

11. Make additional table and carry out follow-up comparison of those HoFH patient having transplantation and those who didn’ Discuss the prognosis both cases. Where there attempts to improve treatment among those who didn’t manage to get liver transplantation?

Response: We quite agree with you that it is of great interest to conduct an RCT trial to compare LT with non-LT treatment for HoFH. However, LT is generally used as a treatment of last resort when all existing treatment strategies have failed; therefore, retrospective comparisons of LT with non-LT for HoFH are not comparable. In addition, most patients are not under our care until they are admitted to our hospital for LT, so please understand that we cannot provide attempts to improve treatment among those who didn’t manage to get LT.

12. Figure 1. I think that readers are interested to get exact TC and LDL values. Provide table or make them otherwise available in the figure.Figure 2. Provide also respective values for non-operated patients. Authors may like to know how fast LDL did decrease? Additional tip: Because the journal is online you can use colours in figures. This will clarify them. 

Response: Thanks for your constructive suggestion, which is extremely valuable for improving our manuscript. As you requested, we have provided detailed baseline, post-LT and lastest TC and LDL values in Table 2 of the revised manuscript. Furthermore, the original Figure 1 has been removed because all the relevant information is already provided in Table 2.

13. Language: some expressions are not so commonly used. Check the language.

Response: We are very sorry for the inappropriate expressions in this manuscript and inconvenience they caused in your reading. Previously, this manuscript has been submitted to Journal of Clinical Medicine and has been edited by the Language Editing Services of MDPI. Since we have made extensive revisions and rewritten the manuscript according to your comments, the manuscript has been further edited by American Journal Experts (AJE).

14. I noticed that authors need to carry out carefully literature review and references. see also my earlier comments. Below is just as an example one important reference which is missing. Additional references: Gao M, Yu W, Hu H, Liu H, Fan K, Gu C, Wang L, Yu Y. Case Report: Cardiac Surgery and Combined Lipid-Lowering Drug Therapy for Homozygous Familial Hypercholesterolemia. Front Pediatr. 2020 Oct 22;8:535949.

Response: All the relevant references have been carefully read, thoroughly discussed, and fully cited in the revised manuscript (Reference # 49 and others).

Reviewer 2 Report

This is a case study series of 12 homozygous FH children from China, in whom 7 had undergone LT (1 at another hospital) and the remaining are awaiting LT.

LT is generally considered a last resort for HoFH patients. Aggressive lipid-lowering therapy indicated for HoFH is generally high-intensity statins, ezetimibe, PCSK9-directed therapy, lomitapide, ANGPTL3-directed therapy and/or lipoprotein apheresis. Most of these patients were not on guideline recommended aggressive lipid-lowering therapy, can the authors please explain why? Some were just on statins alone or just statins + ezetimibe. Also what are the local guideline recommendations of Xuezhikang for HoFH?

Figure 1, the inclusion of those who did not receive LT was not informative. Suggest to just show the 6 cases that had undergone LT.

Author Response

Replies to Reviewer 2

First of all, we would like to express our sincere gratitude to you for your constructive and useful comments on our manuscript.

1. LT is generally considered a last resort for HoFH patients.Aggressive lipid-lowering therapy indicated for HoFH is generally high-intensity statins, ezetimibe, PCSK9-directed therapy, lomitapide, ANGPTL3-directed therapy and/or lipoprotein apheresis.

Response: Thank you for your insightful suggestion. Indeed, LT should be used as a last resort for HoFH patients, and aggressive lipid-lowering therapy should be practiced first. Accordingly, we have discussed the advantages and limitations of various lipid-lowering medications in the Introduction and Discussion sections of the revised manuscript (Line 52–57, 231–243).

2. Most of these patients were not on guideline recommended aggressive lipid-lowering therapy, can the authors please explain why? Some were just on statins alone or just statins + ezetimibe. Also what are the local guideline recommendations of Xuezhikang for HoFH?

Response: We are extremely grateful to reviewer for pointing this out. It should be noted that only patients #1 and #2, who underwent LT in 2014 and 2015, were on statins alone. At that time, ezetimibe was not licenced in our country, and the use of statin was the main option of the medical therapy for HoFH. Patients #3, #4, #5, and #6 underwent LT after 2018, and the use of statins with ezetimibe was the cornerstone of the medical therapy for HoFH and was recommended by the national guidelines for the diagnosis and treatment of rare diseases in China. Despite of this, it needs to become clear that xuezhikang is not recommended as a part of treatment by the aforementioned guidelines. Furthermore, the availability of novel lipid-lowering agents, high costs, and drug reactions would have limited the implementation of guideline recommended aggressive lipid-lowering therapy.

3. Figure 1, the inclusion of those who did not receive LT was not informative. Suggest to just show the 6 cases that had undergone LT.

Response: We agree with your comment that only patients with HoFH who had undergone LT should be included because the topic of this manuscript is LT for HoFH. Since the changes in LDL and TC for each patient have been shown in Table 2 of the revised manuscript, Figure 1 was finally removed.

Reviewer 3 Report

The manuscript covers an important gap in knowledge and is well written. I have a few comments:

-The age range is quite wide in the discussed cases. Could you discuss this  in more detail? What factor may age play in consideration for LT, and what is the impact on clinical outcomes?

-Since your main topic is therapy of FH, you should at least mention the newer treatment options in FH like evinacumab etc and their impact on future FH therapy. It can be expected that thanks to these new therpeutic options some patients might not need LA anymore or intervals can be widened. In this respect, LT will probably not be necessary anymore - please discuss.

-Conclusion section line 315-316: "Our  data suggest that pediatric HoFH patients can undergo LT successfully with an excellent prognosis after careful preoperative assessments and cardiac precautions" - The conclusion section should be rephrased / weakend. LT requries lifelong immunosuppressive therapy and is thus a limited therapeutic option  only for selected patients. Also new drugs will provide more effective lipid-lowering in the near future.

Author Response

Replies to Reviewer 3

First of all, we would like to express our sincere gratitude to you for your constructive, encouraging and positive comments on our manuscript.

1. The age range is quite wide in the discussed cases. Could you discuss thisin more detail? What factor may age play in consideration for LT, and what is the impact on clinical outcomes?

Response: We are grateful for your insightful suggestion. Since the risk of ASCVDs in HoFH patients increases with age and decreases with decreasing LDL-C levels. The older the patient, the worse the treatment effect of LT, and the greater the risks of perioperative major adverse cardiovascular events and ASCVD progression. Accordingly, we have discussed about these topics in the Introduction and Discussion sections of the revised manuscript (Line 47–51, 263–275).

2. Since your main topic is therapy of FH, you should at least mention the newer treatment options in FH like evinacumab etc and their impact on future FH therapy. It can be expected that thanks to these new therpeutic options some patients might not need LA anymore or intervals can be widened. In this respect, LT will probably not be necessary anymore - please discuss.

Response: Thank you for pointing out this important issue. We have summarized the advantages and limitations of various lipid-lowering drugs, LDL apheresis, gene therapy, hepatocyte transplantation and liver transplantation in the treatment of HoFH in detail in the Introduction and Discussion sections of the revised manuscript (Line 52–61, 231–256).

3. Conclusion section line 315-316: "Ourdata suggest that pediatric HoFH patients can undergo LT successfully with an excellent prognosis after careful preoperative assessments and cardiac precautions" - The conclusion section should be rephrased / weakend. LT requries lifelong immunosuppressive therapy and is thus a limited therapeutic option only for selected patients. Also new drugs will provide more effective lipid-lowering in the near future.

Response: As you requested, we have modified the inappropriate descriptions in the Abstract and Conclusion sections of the revised manuscript (Line 52–61, 231–256).

Round 2

Reviewer 1 Report

I thank you for your comments. I ask one correction . Regarding the table mentioning "*Xuezhikang is a traditional Chinese medicine with lipid-lowering effects." Mention that this medication is not recommended according to current guidelines.

Author Response

Replies to Reviewer 1

First of all, we would like to express our sincere gratitude to you for your constructive and useful comments on our manuscript.

1. Regarding the table mentioning "*Xuezhikang is a traditional Chinese medicine with lipid-lowering effects." Mention that this medication is not recommended according to current guidelines.

Response: We are extremely grateful to reviewer for pointing this out. Indeed, Xuezhikang is not recommended as a part of treatment by the current treatment guidelines; therefore, we have removed this non-standard treatment option from Table 2 of the revised manuscript (Line 173).